# Effect of a Mismatched Vaccine against the Outbreak of a Novel FMD Strain in a Pig Population

**DOI:** 10.3390/ani13193082

**Published:** 2023-10-02

**Authors:** Jun-Hee Han, Dae-Sung Yoo, Chang-Min Lee

**Affiliations:** 1EpiCentre, School of Veterinary Science, Massey University, Palmerston North 4410, New Zealand; j.h.han@massey.ac.nz; 2College of Veterinary Medicine, Chonnam National University, Gwangju 61186, Republic of Korea; shanuar@jnu.ac.kr

**Keywords:** foot and mouth disease, unmatched vaccine, antigenic diversity, vaccine effect, ABC-SMC

## Abstract

**Simple Summary:**

An outbreak of foot-and-mouth disease from a pig population vaccinated with a mismatched strain raises a question about how effective a mismatched foot and mouth vaccine could be against the transmission of another strain of the virus. Simulation models using daily clinical case data from the farm indicated that, although the vaccine reduced viral shedding, it failed to prevent the development of clinical signs. With caveats, this study suggests that using unmatched vaccines could alter the disease dynamics of another strain and, in turn, may confound the outbreak in a vaccinated population. It is, therefore, important to test for antigenic matching between circulating strains and the vaccine pool.

**Abstract:**

In December 2014, a novel foot and mouth disease (FMD) virus was introduced to a pig farm in South Korea, despite the animals being vaccinated. A marginal antigenic matching between the novel and vaccine strains potentially led to the infection of the vaccinated animals. To understand the impact of using an FMD vaccine on the transmission dynamics of an unmatched field strain, simulation models were employed using daily reported data on clinical cases from the farm. The results of this study indicated that immunisation with the FMD vaccine reduced the shedding of the novel FMD virus in pigs. However, there was no evidence to suggest that the immunisation had a significant effect in reducing the development of clinical signs. These findings highlight that the use of an unmatched FMD vaccine can confound the outbreak by altering the disease dynamics of the novel virus. Based on this study, we emphasise the importance of continuous testing to ensure antigenic matching between the circulating strains and the vaccine pool.

## 1. Introduction

Foot and mouth disease (FMD) is a contagious disease in cloven-hoofed animals [1]. Depending on the host species or FMD virus strain, infected domestic livestock would show oral and podal vesicular lesions causing a reduction in production due to pain [2]. FMD outbreaks and efforts to control the disease could often cause an enormous economic burden [3] and FMD vaccination has been conducted in many endemic countries as a cost-effective control method [4]. Even with vaccination, however, a vaccine failure could occur, potentially due to the poor vaccine quality and/or poor vaccination planning [5,6]. In occasion, FMD vaccinations fail to prevent FMD outbreaks due to the emergence of a new variant of the virus causing an antigenic mismatch between the vaccine and novel strains [7,8,9].

In case of an FMD outbreak in a population with FMD vaccination of an unmatched strain, the vaccination may still confer a non-negligible level of immunity and affect the disease dynamics in the population. For example, depending on the variation in antigenicity and vaccine quality, vaccinated animals might still be protected from infection at a certain level when they are exposed to heterologous FMD virus, shed less amount of virus, or be less likely to develop clinical signs when they are infected with the virus [10,11]. The extent to which disease dynamics are altered might vary based on the immunogenic similarities between the field and vaccine strains as well as the quality of vaccines. However, the magnitude of such alterations could be of interest in some countries with ongoing FMD vaccination, because it may increase the risk of initial transmission of a novel field strain due to the lack of obvious clinical signs in some immunised animals or misguided belief that vaccination would be effective.

After the epidemics of FMD in 2010/2011, the government of the Republic of Korea adopted a mandatory vaccination policy covering all cloven-hooved domestic animals to prevent the widespread FMD outbreaks against any future introduction [12]. The policy seemed to be effective as no outbreaks were reported until July 2014, when FMD virus (e.g., O/SKR/Jul/2014) was introduced, but its impact was limited to only three pig farms [13]. However, when another strain of FMD virus (O/Jincheon/SKR/2014) was introduced to the country in December 2014, the disease managed to spread to a total of 180 pig and five beef farms over a span of five months, resulting in an estimated economic loss of approximately USD 25 million [14]. Subsequent investigation revealed an antigenic variation between the vaccine strain (O1 Manisa) and novel field strain (O/Jincheon/SKR/2014) [15], raising questions about the effectiveness of the vaccination policy with the particular strain in preventing the transmission of the novel FMD virus.

To address such questions, we conducted a study using data from a commercial farm that experienced an FMD outbreak in 2014. Our primary objective was to examine the impact of using a FMD vaccine (O1 Manisa) on the transmission dynamics of an unmatched field strain (O/Jincheon/SKR/2014) in a pig population. Specifically, we aimed to assess the effect of immunisation with the vaccine strain in reducing the transmission and development of clinical signs of the novel FMD strain by employing an approximate Bayesian computation method.

## 2. Materials and Methods

### 2.1. Epidemiological Context

An FMD outbreak was reported from a commercial pig farm in Jincheon, South Korea on 3rd December 2014, with four sows showing vesicular lesions on the snout and lameness caused by detachment of the claw horn. In the next few days, sows in the same barn and other pigs in different barns also showed similar clinical signs. Given the mandatory FMD vaccination policy of the country, local veterinary officers surveyed the farm owner and collected blood samples of random pigs in the farm on 6th December to conduct FMD Structural Protein (SP) antibody ELISA (PrioCHECK™ FMDV Type O Antibody ELISA, Thermo Fisher Scientific, Waltham, MA, USA) to examine the compliance of the farm owner to the policy. The survey and ELISA results confirmed that the animals were routinely vaccinated (e.g., FMD vaccination at 8 weeks old for piglets and 4 weeks before farrowing for sows) and, thus, the officers and farm owner decided to cull only the pigs showing the clinical signs (i.e., clinical FMD) while other pigs yet to be vaccinated at the time of testing were pre-emptively culled. Nevertheless, the incidence of clinical FMD continued over the following weeks and, in turn, all the animals in the farm were mandated to be culled on 17th December (Figure 1).

### 2.2. Data

In the farm, there were six nursery barns with vaccinated pigs exempted from pre-emptive culling in early December. Each barn comprised a different number of pig pens, housing a total of approximately 400 or 800 nursery pigs aged between 8 and 10 weeks. During the outbreak, the nursery pigs were visually examined twice a day and removed immediately if the pigs showed any FMD-specific clinical signs. The first clinical FMD among the nursery pigs was observed on 6th December (i.e., Day 1) and the number of removed pigs from each nursery barn was recorded until 16th December (i.e., Day 11). Among the nursery pigs, only the pigs from four barns were sampled for FMD SP antibody ELISA conducted on December 6th. Given the proportion of animals with passive immunity (i.e., immunised pigs, hereafter), conferred by either the vaccination or the intake of colostrum from vaccinated sows was the key information of this study, we extracted the record from the four barns for further processing. The description of the extracted record is illustrated in the Appendix A).

### 2.3. Modelling Approach

Given the lack of knowledge around disease dynamics of a novel FMD virus in pigs immunised with a mismatched strain, we established five FMD models (i.e., $M_{1},M_{2},M_{3},M_{4},$ and $M_{5}$), each of which represents a different transition process between disease status. We then compared the model fitness to the observed data (i.e., record of removed animals per day) using an approximate Bayesian computation–sequential Monte Carlo (ABC-SMC) algorithm and selected one that generated the best description of the data. Once the best model is selected, the values of parameters related to the disease dynamics were inferred by re-running the ABC-SMC algorithm separately for the model. The disease simulations were implemented in the C programming language and the ABC-SMC algorithm was run in R [16].

#### 2.3.1. Brief Model Description

Nursery pigs in the models were initially categorised as “susceptible” to the novel FMD strain, and once infected, the disease status of the animals transited to “exposed”. After a latent period ($\gamma_{1}$), infected pigs became “sub-clinically infectious” and shed the virus for $\gamma_{2}$ days (i.e., sub-clinically infectious period). After the sub-clinically infectious period, the infected animals eventually developed clinical signs, by which time the pigs were removed from the barns to mimic the observed record.

In $M_{1}$, the immunisation was assumed to have no preventive effect against the infection of the novel FMD strain, so that immunised nursery pigs were treated the same as susceptible pigs. Immunised pigs were also susceptible to the novel strain in $M_{2},M_{3},\;and\;M_{4}$; however, the course of FMD infection for the immunised pigs was modified by incorporating one or two epidemiologic features; the development of clinical FMD was partially prevented in $M_{2}$, FMD transmission was reduced in $M_{3}$, and both features were included in $M_{4}$. Lastly, immunised pigs in $M_{5}$ were assumed to be fully protected from infection. A concise description of the five models is provided in Table 1, and a detailed description of each simulation model, including the definition of model parameters, is provided in the Appendix A.

In the models, we assumed that the proportion of positive results to FMD SP antibody ELISA indicates the proportion of pigs that acquired a proper level of immunisation (to the vaccine strain) in each barn (except in $M_{1}$). All the models were stochastic density-dependent individual-based models with a day as a temporal unit. The simulation period was 20 days: from Day −8 (i.e., the earliest day of novel FMD virus introduction, which was assumed to be nine days earlier from Day 1 [17]) to Day 11 (i.e., the last day of the removal record). Given the short simulation period, natural mortality or waning of passive immunity was not considered in the model.

#### 2.3.2. Model Selection and Parameter Estimation

To select the best model and estimate the values of related parameters, we used an ABC-SMC algorithm [18]. In brief, the algorithm infers parameters through a series of estimation sequences, each of which has a decreased distance threshold than its previous sequence. In each sequence, parameter values are randomly sampled based on the perturbation of the values in the previous sequence (i.e., perturbation kernel). The sampled values are then applied to a model to generate simulated data, of which the distance of summary statistics to the one of observed data is measured. Only the parameter values that generate the simulated data with a distance less than a threshold of the current sequence are retained until a number of values (i.e., particle) are collected, and the whole process repeats until the particles of the final SMC sequence are collected. The distribution drawn by the final particles of a parameter is a marginal posterior distribution of the parameter, which approximates the posterior distribution of the parameter given the observed data. A benefit of using the ABC-SMC algorithm is that the posterior distribution of parameters can be inferred even when the likelihood of the model is intractable [19]. The algorithm also can be extended to compare the fitness between different models by conducting an additional step of the random model selection process before sampling parameter values for a particle of each sequence. This additional step makes a model with a better description of observed data (i.e., model generates simulated data with a lesser distance of summary statistics given the sampled parameter values) more likely to be retained. A more detailed explanation of parameter estimation and model selection process using ABC-SMC is in [18]. The detailed algorithm for the ABC-SMC used in this study is provided in the Appendix A.

We assumed that all nursery pigs in the four barns shared the common distribution of $\alpha ,\beta ,\gamma_{1},\gamma_{2},\gamma_{3},\varphi ,$ and ω while other parameters ($\rho_{i}$ and $\tau_{i}$) varied between the barns (Table 2). The prior distribution of the within-pen transmission rate (β) was assumed to be normally distributed with the mean and standard deviation of 0.6 and 0.1 per day, respectively [20], and the distribution was truncated within the minimum and maximum value of 0.0 and 1.5 per day [21], respectively. The reduced infectious pressure between pigs in different pens was adjusted by incorporating the relative decrease in β for between-pen transmission (ω) with its prior of a uniform distribution between 0 and 0.5 [22]. The prior distribution of the latent period ($\gamma_{1}$) and sub-clinically infectious period ($\gamma_{2}$) followed binomial distributions with the sum of $\gamma_{1}$ and $\gamma_{2}$ to be ≤ 9 days [17], while the prior of the sub-clinically infectious period for immunised pigs before recovery ($\gamma_{3}$) was a uniform distribution between 3 and 7 days [10]. The prior distribution of the proportion of immunised pigs ($\rho_{i}$) was a beta distribution with the shape parameters being the number of FMD SP antibody positive and negative samples from barn *i*. The prior of the day of novel FMD virus introduction ($\tau_{i}$) was a uniform distribution between −8 and 11, with −8 indicating the introduction of the novel FMD virus was on Day 8 (i.e., 27th November) to barn *i*. Given the lack of information, uniform priors between 0 and 1 were used for the remaining parameters.

For the perturbation kernel, a component-wise Gaussian kernel with the variance as twice the variance of particles in the previous sequence was used [23]. The summary statistics of this study was identical to the observed data, which was the number of removed nursery pigs (due to the development of clinical signs) per day per barn. The distance ($D_{i}$) between the simulated and observed data for barn *i* was calculated as
$$D_{i}=\sqrt{\overset{T}{\underset{t=1}{\sum }}{\left(S_{t}^{i}-O_{t}^{i}\right)}^{2}}$$
where T is the duration that the pigs with clinical signs were removed (i.e., 11 days; from 6th to 16th December), and $S_{t}^{i}$ and $O_{t}^{i}$, respectively, indicate the simulated and observed number of pigs that were removed due to the development of clinical signs at Day *t* from barn *i*.

For model selection, we set the ABC-SMC algorithm to accept 500 particles and reduce the distance thresholds of the subsequent sequence as the median of the distances generated by the accepted particles from the current sequence. After repeating the algorithm for 12 sequences, a model that best described the observed data was selected using the Bayes factor. The Bayes factor was calculated as a pairwise division of the number of models being selected in the final sequence [18]. We assumed that the value of the Bayes factor > 3 was evidence of one model showing a better fit to the data over another one [24]. Once a model was selected, we repeated the ABC-SMC algorithm separately for the selected model to infer the parameter values.

### 2.4. Validation and Sensitivity Analysis

The selected model and the marginal posterior distribution of its parameters were validated by reproducing the transmission of the novel FMD virus in the nursery pigs of the four barns. To do so, the model was iterated 500 times using the parameter values from the final sequence, and we visually examined whether $O_{t}^{i}$ was within a 95% prediction interval of $S_{t}^{i}$ from the iteration.

We also conducted sensitivity analyses for the selected model to investigate the following: (i) Which parameter most affected the summary statistics as well as the total number of pigs developing clinical signs; (ii) What would be the impact of the lack of prior information for $\beta ,\gamma_{1}$, or $\gamma_{2}$ in inferred parameter values. In the first sensitivity analysis, we examined eight parameters (e.g., $\beta ,\gamma_{1},\gamma_{2},\varphi ,\omega ,\rho ,\tau$ and the number of pigs initially exposed to the novel FMD) by randomly and independently selecting each parameter value from either its prior distribution (in case of $\beta ,\gamma_{1},\gamma_{2},\varphi ,\omega ,\rho ,$ and τ) or between 1 and 5 (in case of the number of initially exposed pigs) and incorporating this into the selected model. We iterated this process 10,000 times, and the partial rank correlation coefficient (PRCC) of the parameters was evaluated for each of the computed outcomes [25,26]. The second sensitivity analysis was identical to the parameter estimation process above except for changing the prior distribution of either $\beta ,\gamma_{1}$, or $\gamma_{2}$ to a uniform distribution.

## 3. Results

### 3.1. Model Selection

Figure 2 illustrates the number of five models being retained over 12 sequences. The most retained model in the last sequence was $M_{3}$ with 133 out of 500 times being selected, followed by $M_{4}$ (43 times), $M_{1}$ (13 times), and $M_{2}$ (9 times). This resulted in the Bayes factors of $M_{3}$ compared with other models to be between 3.1 and 66.5 and, therefore, $M_{3}$ was selected for the inferring process of parameter values.

### 3.2. Parameter Values

The posterior distribution of parameters relevant to $M_{3}$ is provided in Table 3. The median of the within-pen transmission rate (β) and the relative decrease in β for between-pen transmission (ω) was 0.27 and 0.004, respectively. The median value of the reduction in viral shedding of immunised pigs (φ) was approximately 0.4. The median of both the latent period ($\gamma_{1}$) and sub-clinically infectious period ($\gamma_{2}$) was 2 days. The posterior distributions of the whole estimated parameters are provided in the Appendix A.

### 3.3. Validation

For comparison with the observed data, the number of pigs removed due to the development of clinical signs over the simulation period for each barn was simulated by using the estimated parameter values to $M_{3}$ (Figure 3). Except for barn 2, the observed data was generally within the 95% prediction interval of the simulated data.

The sensitivity analysis based on PRCC showed that, in $M_{3}$, both the summary statistics and accumulative number of clinical FMD were most affected by the day of novel FMD virus introduction (τ), followed by the latent period ($\gamma_{1}$). The posterior distributions of the parameters were highly affected by a prior distribution of $\gamma_{2}$; however, no distinct differences were observed in the posterior distributions when a uniform prior of either β or $\gamma_{1}$ was used. Detailed results of the sensitivity analysis are provided in the Appendix A.

## 4. Discussion

Vaccination against FMD is widely recognised as a cost-effective strategy to combat its highly transmissible nature and mitigate economic losses [27]. To ensure optimal protection, it is crucial to monitor the antigenic compatibility between the vaccine and circulating strains, thus avoiding potential vaccine failures [8]. However, given the inherent mutability of the FMD virus combined with the escalating levels of international travel and trade, the risk of introducing novel viral strains not covered by the administered vaccines could be a pressing concern. Therefore, this study offers valuable insights that shed light on the potential consequences associated with the utilisation of mismatched vaccines, serving as an informative resource for decision making.

To better understand the effect of FMD immunisation with the O1 Manisa strain on the transmission dynamics of the O/Jincheon/SKR/2014 strain in a pig population, we developed five simulation models, each representing distinct transmission dynamics. These models were then compared to assess the fitness to the observed data. This study showed that the model assuming the reduced FMD transmission of immunised pigs (e.g., $M_{3}$) demonstrated the closest fit to the data, suggesting that immunisation with the mismatched vaccine strain had a certain degree of inhibitory effect on the viral shedding of the field strain. The fact that $M_{3}$ outperformed the rest of the models implies two important points. Firstly, immunisation with the O1 Manisa strain had some protective effects against the FMD outbreak in December 2014, South Korea (against $M_{1}$). Secondly, although it may help reduce the infectious pressure, it did not prevent immunised pigs from developing clinical signs (against $M_{2}$ and $M_{4}$). This would be in line with [15], who implied that the expected level of vaccine protection was marginal based on the matching value (or so-called $r_{1}$-value) of 0.1~0.3 (VNT test). Interestingly, the prior outbreak in July 2014 was well controlled with the same vaccine strain and vaccination policy (e.g., vaccine schedule or dose), even though the $r_{1}$-value between the vaccine strain and the strain introduced in July 2014 (O/SKR/Jul/2014) was only 0.14 [28]. This suggests that the $r_{1}$-value cannot be used as a sole indicator to estimate the efficacy of cross-protection via FMD vaccination. Because the cross-protection of FMD vaccine against a heterologous virus strain would rely on various components, such as the antigenic variability, vaccine potency, or booster doses [29], it is important to note that such an alteration in the disease dynamics could be specific to the O1 Manisa and O/Jincheon/SKR/2014 strains and, thus, may not be applicable to the strains with the similar $r_{1}$-value.

The within-pen transmission rate (β) was estimated as 0.27 per day, with a 95% credible interval ranging from 0.07 to 0.52 per day. This estimate is notably lower than the prior distribution (e.g., normal distribution with the mean and standard deviation of 0.6 and 0.1, respectively) re-calculated from [20]. However, it is important to consider that the distribution of β reported by [20] represents the entire infectious period, whereas the estimated β in our study only pertains to transmission during the sub-clinically infectious phase of pigs. Previously, it was demonstrated that pigs exhibiting clinical signs shed a larger quantity of FMD virus compared with asymptomatic or pre-clinical pigs [10]. Consequently, the lower value of the posterior distribution of β compared to its prior would be explained by this observed difference in shedding dynamics.

The median of the relative decrease in β for between-pen transmission (ω) was 0.004, indicating that the novel FMD virus slowly spread across different pens inside barns of the outbreak farm. Similar to our findings, a substantial reduction in FMD transmission rate caused by the physical barrier such as pen walls has been reported in pigs. For example, it was experimentally demonstrated that the transmission rate of FMD between pens was approximately 10 times lower than the rate within a pen for non-vaccinated pigs [30]. Similarly, a 20-fold reduction in FMD transmission in a non-vaccinated pig population was reported due to the presence of pen walls [31]. Additionally, the authors of [31] indicated that no between-pen transmission was observed when the distance between pens exceeded 40cm. Considering the low ω value observed in our study, as well as the findings from the aforementioned literature, we believe that the infectious pressure across pens within the outbreak farm was minimal.

The estimated relative decrease in β for immunised pigs (φ) was approximately 0.4 (95% credible interval of 0.03~0.94). This potentially implies that, when infected, nursery pigs immunised with the vaccine strain shed about 40% less of the novel FMD virus compared to non-immunised pigs. It has been demonstrated that FMD vaccination could significantly decrease the amount of live virus excretion in pigs when they are infected with a semi-heterologous strain [10]. In another study based on meta-analysis, the within-pen transmission rate for non-vaccinated pigs was calculated to be approximately nine times higher than the rate for pigs vaccinated with heterologous vaccine [20]. While the exact impact may vary depending on the level of antigenic similarity, vaccination schedule, or vaccine doses [21,32], our results support the previous findings that the application of a heterologous vaccine can reduce the shedding of FMD virus in pigs.

Conclusively, we strongly believe that the combination of aforementioned factors (e.g., limited within-pen transmission caused by the immediate removal of clinical cases, limited transmission between pens, and reduced viral shedding of immunised pigs) collectively contributed to a slowed transmission of the novel FMD strain within the outbreak farm. This contrasted with the expected high transmissibility observed during FMD epidemics in 2010/2011 [12]. Along with the experience of effective FMD control using the same vaccine strain in July 2014 [13], the slower transmission dynamics observed in this case could have potentially perplexed veterinary officials, creating a false perception that vaccination was somewhat effective. Consequently, it delayed the prompt implementation of crucial administrative measures, such as standstill protocols and proactive pre-emptive culling, by the central government, resulting in the widespread transmission of the novel strain [14]. Although speculated, our re-constructed reflection provides a valuable but obvious lesson—FMD vaccine failure can occur regardless of the effectiveness of the vaccination in previous outbreaks. This emphasises the importance of continuous monitoring of novel FMD strains and rigorous testing for antigenic matching with the vaccine pool [8].

As with any research based on simulation modelling, our study does have certain limitations that should be acknowledged. Firstly, the inferred parameter values were found to be sensitive to the prior information regarding the sub-clinically infectious period. This is not surprising, considering that sub-clinically infectious pigs were the sole source of infection in the model. Consequently, the identifiability of the model heavily relied on the accuracy of the information concerning the duration of viral shedding. In our study, the prior distribution of the sub-clinically infectious period was based on [17], where the distribution was estimated through a meta-analysis of the literature. It is worth noting that the estimation predominantly focused on the FMD O serotype and the novel FMD strain in this study also belonged to this serotype. Therefore, it would be less likely that the actual sub-clinically infectious period of the O/Jincheon/SKR/2014 strain in pigs significantly deviated from the prior distribution.

Another limitation of this study would be a model assumption that only a proportion of pigs acquired a proper level of immunity against the vaccine strain even though all the pigs were vaccinated. The FMD vaccine programme in the outbreak farm started at 60 days old, and the pigs modelled in this study were between 8 and 10 weeks old. This suggests that some animals might not have attained proper immunisation due to the relatively short period from the vaccine administration with the absence of booster as well as the loss of or interference with maternally derived antibodies [33]. To account for this, we, therefore, incorporated the proportion of SP ELISA-positive pigs into the model as a proxy of the proportion of properly immunised pigs. However, we acknowledge that assuming the pigs were classified as being immunised solely based on the SP ELISA positivity would be an oversimplification of the complex immune response.

Several studies have been conducted to estimate key parameter values related to FMD transmission [20,21,30,31]. These estimations typically involve fitting observed data from controlled experimental settings to simulation models. In contrast to controlled experiments, where researchers can meticulously monitor the development of FMD clinical signs and measure viral shedding, our study draws from observations made in a real-world commercial farm setting. Although farm workers diligently examined the pigs twice a day, it is essential to acknowledge that, in such observational settings, there is a possibility that some pigs displaying clinical signs of FMD may have been missed and not promptly collected and removed. We acknowledge that the potential for this type of bias is an inherent aspect of observational studies that rely on passively collected data, as is the case with our research. It is, therefore, important to consider such limitations when interpreting the results of this study.

The simulated data generated based on the posterior distribution of the parameters did not completely align with the observed data from barn 2. There are a couple of potential explanations for this discrepancy. Firstly, it is possible that there were more than 50 pigs present in the pen where the novel strain was initially introduced in the barn. This could have affected the dynamics of the transmission within the pen and led to variations in the observed data. Another possible explanation is that more than one pig was initially exposed to the virus in the barn, resulting in a stronger infectious pressure than what was assumed in the simulation. However, due to the lack of available evidence or records from the outbreak, we can only speculate on these potential explanations. It is important to note that uncertainties and limitations exist in any modelling study, and without concrete evidence, our speculations remain speculative. Further research and more comprehensive data would be necessary to gain a clearer understanding of the factors influencing the observed data; however, it will be a difficult task given the age of the outbreak.

## 5. Conclusions

Based on the outbreak data from a commercial pig farm in South Korea, our investigation focused on assessing the impact of an unmatched FMD vaccine against the infection and transmission of a novel FMD strain. It was concluded that immunised pigs shed a lesser amount of the novel virus when they were infected, which contributed to a reduction in the infectious pressure and slowed down the spread of FMD within the farm. It is likely that such alterations in the disease dynamics confounded veterinary officials to perceive that the vaccine is somewhat effective. This emphasises the importance of testing for antigenic matching between the field FMD strains and the vaccine pool.

## Figures and Tables

**Figure 1 animals-13-03082-f001:**
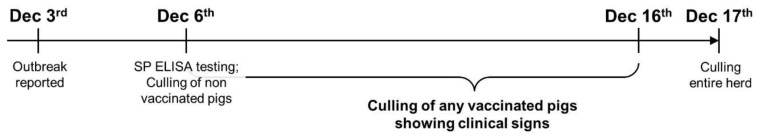
A timeline of foot and mouth disease outbreak response at a pig farm in South Korea (2014).

**Figure 2 animals-13-03082-f002:**
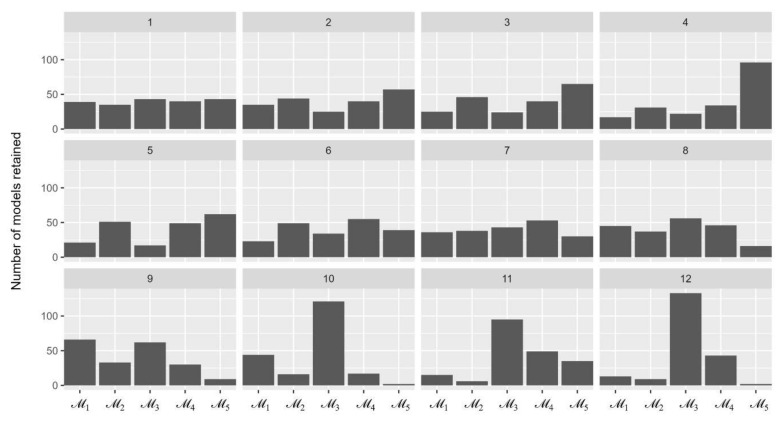
The number of simulation models remained over 12 sequences of the approximate Bayesian computation process.

**Figure 3 animals-13-03082-f003:**
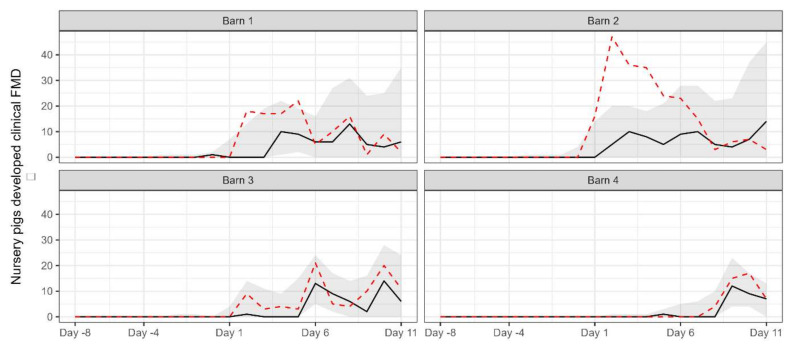
The median (black solid line) and 95% prediction interval (grey area) of the simulated number of pigs removed due to the development of clinical signs for each barn. The red dashed line indicates the observed number of removed pigs.

**Table 1 animals-13-03082-t001:** The brief description of five simulation models in this study.

Incorporated Features	$M_{1}$	$M_{2}$	$M_{3}$	$M_{4}$	$M_{5}$
*Transmission dynamics*					
Immunised pigs less likely develop clinical signs	-	Yes	-	Yes	-
Immunised pigs less likely transmit disease	-	-	Yes	Yes	-
Immunised pigs exempted from infection	-	-	-	-	Yes
*Parameters*					
Within-pen transmission rate (*β*)	Yes	Yes	Yes	Yes	Yes
Relative decrease in *β* for between-pen transmission (*ω*)	Yes	Yes	Yes	Yes	Yes
Latent period ($\gamma_{1}$)	Yes	Yes	Yes	Yes	Yes
Sub-clinically infectious period before developing clinical signs ($\gamma_{2}$)	Yes	Yes	Yes	Yes	Yes
Sub-clinically infectious period for immunised pigs before recovery ($\gamma_{3}$)	-	Yes	-	Yes	-
Relative decrease in β for immunised pigs (φ)	-	-	Yes	Yes	-
Proportion of immunised but sub-clinically infectious pigs to recover (α)	-	Yes	-	Yes	-
Proportion of immunised pigs in barn *i* ($\rho_{i}$)	-	Yes	Yes	Yes	Yes
Proportion of immunised pigs in barn *i* ($\rho_{i}$)	Yes	Yes	Yes	Yes	Yes

**Table 2 animals-13-03082-t002:** The description and prior distribution of parameters examined for five models in this study.

Parameters	Prior	References
Within-pen transmission rate (β)	*N* (0.6, 0.1^2^) per day 0 ≤ β ≤ 1.5	Calculated from [20,21]
Relative decrease in β for between-pen transmission (ω)	*U* (0.0, 0.5)	[22]
Latent period $(\gamma_{1}$) *	*Binomial* (97, 0.02) day	[17]
Sub-clinically infectious period before developing clinical signs $(\gamma_{2}$) *	*Binomial* (66, 0.02) day	[17]
Sub-clinically infectious period for immunised pigs before recovery $(\gamma_{3}$)	*U* (3, 7) day	[10]
Relative decrease in β for immunised pigs (φ)	*U* (0, 1)	No info.
Proportion of immunised but sub-clinically infectious pigs to recover (α)	*U* (0, 1)	No info.
Proportion of immunised pigs $(\rho_{i}$) ^†^	*Beta* (a, b) ^‡^	Assumed
Day of novel FMD virus introduction $(\tau_{i}$) ^†^	*U* (−8, 11)	Assumed

Key: *U*, uniform distribution; *N*, normal distribution. * The sum of $\gamma_{1}$ and $\gamma_{2}$ was assumed to be ≤ 9 days. ^†^ The value varied between nursery pig barns. ^‡^ The values a and b were the number of samples positive and negative to FMD SP antibody ELISA in a barn, respectively.

**Table 3 animals-13-03082-t003:** The posterior distribution of the estimated parameters in $M_{3}$.

Parameter	Median (95% Credible Interval)
Within-pen transmission rate (β)	0.269 (0.066, 0.524) per day
Relative decrease in β for between-pen transmission (ω)	0.004 (0.000, 0.019)
Latent period ($\gamma_{1}$)	2 (1, 2) days
Sub-clinically infectious period before developing clinical signs ($\gamma_{2}$)	2 (1, 3) days
Relative decrease in β for immunised pigs (φ)	0.401 (0.031, 0.941)

## Data Availability

The source codes of the models and ABC-SMC algorithm are freely accessible at https://github.com/junhhan/FMD_Korea.

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
