# Peer review of "Effect of a Mismatched Vaccine against the Outbreak of a Novel FMD Strain in a Pig Population"

_animals, 2023, doi:10.3390/ani13193082_

Round 1

Reviewer 1 Report

Effect of FMD vaccine against the outbreak of novel strain with a poor antigenic matching in a pig population

Jun-Hee Han, Dae-Sung Yoo, and Changmin Lee

This is an interesting approach, by trying to match modelling results to observations. But the title is then not correct. The authors try to estimate model parameters by matching the model outcomes to the observation. It is very similar to the estimation of transmission rate and infectious period in an experimental transmission study, although in that case there are laboratory data for all pigs present in the study. The approach assumes that the observations were correct, and that no infected pigs were missed and there were no other cases. This assumption can be challenged in the discussion.

The title and text indicates that the lower protection than expected in vaccinated pigs was due to antigenic mismatch, but FMD vaccine induced protection depends mainly on vaccine quality, quality of the vaccination programme and also on antigenic match. In the paper and abstract only antigenic match is mentioned and not the other 2 are more important parameter. The paper is not measuring the antigenic distance, so it should not be mentioned in the title. In every outbreak the outbreak strain does not match the vaccine strain, but good quality vaccines can still protect pigs and reduce transmission.

I can understand the modelling approach, but it is not sufficiently clear described, not in the paper but also not in the supplementary material. For many parameters in table 1 the units are missing. Is the sub-clinical period in minutes or hours?

Comments:

Abstract:

Information on vaccine quality and quality of the vaccination programme is missing in the abstract.

Introduction:

Line 23: delete "highly". FMD is not a highly contagious disease. In endemic regions the R-naught is between 1.3 and 1.8.

Line 25: replace "pedal" by "podal".

Line 26: Initially FMD was economically important only in dairy cattle due to loss of milk production. When it was controlled in dairy cattle, infection in other species became a huge risk to dairy cattle and therefore it became a trade barrier. But losses in pig production would be minimal if there was no trade barrier. Choose a different wording for this sentence, now it is not correct.

Line 28: Most vaccine failures are due to poor vaccine quality and poor vaccination planning (e.g. poor cold chain). Europe used more or less the same vaccine strain for 40 years, this was the same in South America and controlled FMDV. So, antigenic matching is not the main issue for vaccine failure.

Line 33 - 39: the most important issue in FMD vaccination is the reduction of transmission, not the protection of individual animals. Several transmission studies using heterologous vaccine (O Manisa vaccine and O/TAW/97 challenge) show that heterologous vaccination with a good quality vaccine works. These studies should be mentioned in the introduction.

Materials and methods:

Line 77 - 83: Very sloppy to leave the instruction to authors in the paper. This should be deleted.

Line 85 - 87: At which age were the piglets vaccinated? Vaccination in presence of maternally derived antibodies does not work! See:

Francis, M. J., Black, L. 1984. The effect of vaccination regimen on the transfer of foot and mouth disease antibodies from the sow to her piglets. J. Hyg. (Lond). 93(1); 123-131.

Francis, M. J., Black, L. 1986. Response of young pigs to foot-and-mouth disease oil emulsion vaccination in the presence and absence of maternally derived neutralising antibodies. Res. Vet. Sci. 41(1); 33-9.

Kitching, R. P., Alexandersen, S. 2002. Clinical variation in foot and mouth disease: pigs. Rev. Sci. Tech. Off. Int. Epiz. 21(3); 513-518.

Kitching, R. P., Salt, J. S. 1995. The interference by maternally-derived antibody with active immunization of farm animals against foot-and-mouth disease. Br. Vet. J. 151(4); 379-389.

Dekker, A., Chénard, G., Stockhofe, N., Eblé, P. L. 2016. Proper Timing of Foot-and-Mouth Disease Vaccination of Piglets with Maternally Derived Antibodies Will Maximize Expected Protection Levels. Front. Vet. Sci. 3(52).

Line 118 - 124: I suggest to make a table of the 5 models.

Line 158: in reference 16 all estimates of ß are 0.4 - 73, how does the author then assume a normal distribution with mean 0.6 and variance 0.1?

Line 159 - 160 and Table 1: why is a transmission study in sheep [17] used for estimation of the within-pen transmission rate (ß). I do not see anything in reference 17 indicating that 0 ≤ ß ≤ 1.5.

Line 160 - 163: Reference 18 does not mention between-pen transmission estimates. It is given in reference 16, why does the author use an uninformative prior when data are available?

Line 162 - 166: Last year a paper was published on the duration of various periods. Why does the author not use these estimates? Moreno-Torres, K. I., Delgado, A. H., Branan, M. A., Yadav, S., Stenfeldt, C., Arzt, J. 2022. Parameterization of the durations of phases of foot-and-mouth disease in pigs. Prev. Vet. Med. 202; 105615.

Line 178: What is a perturbation kernel. The paper should be readable by interested veterinarians.

Line 170 - 177: There is no definition of "Reduction in viral shedding of immunised pigs (?)", is this a reduction in the duration of infection. The latter is suggested by "Pigs in the Ism status were assumed to be infectious for  days", or do you mean the titre of excreted virus is lower? But how does that fit into the model.

Figure S4 is missing y3, but the text refers to inclusion of y3.

Line 270 - 274: The WRL estimate of the r1-value is published on the WRL website https://www.wrlfmd.org/sites/world/files/quick_media/WRLFMD-2014-00027-SKR-VMR-O-O_001.pdf. A low r1-value does not mean that a vaccine does not work, it will depend on the quality of the vaccine. A vaccine with a homologous potency of 21 PD50/dose vaccine with a r1-value of 0.14 (assuming that the r1-value predicts the potency ratio), the potency against the outbreak strain will still be 3 PD50/dose, which according to OIE is sufficient. So, the measuring vaccine quality is extremely important. All outbreak strains are heterologous to the vaccine unless you use poorly inactivated vaccine.

Line 278 - 279: ß is the average number of new infections caused by a typical infectious individual per day. It is assumed to be equal over the whole infectious period, although you could also scale it based on virus excretion data. But it is not dependent on infection state (clinical or subclinical). What is the definition of the parameters in the current model. This is very unclear for all parameters. Units are missing for all parameters. The unit of ß would be 1/day.

Line 293: Not "wall thickness" but "distance between pens"

Line 300 - 307: Several studies quantifying within-pen transmission using heterologous vaccinestrains have been published. In the Orsel paper there was no significant difference between virus excretion in vaccinated and non-vaccinated pigs if I remember well.

Eblé, P. L., Bouma, A., de Bruin, M. G. M., van Hemert Kluitenberg, F., van Oirschot, J., Dekker, A. 2004. Vaccination of pigs two weeks before infection significantly reduces transmission of foot-and-mouth disease virus. Vaccine 22(11-12); 1372-1378.

Eblé, P., de Koeijer, A., Bouma, A., Stegeman, A., Dekker, A. 2006. Quantification of within- and between-pen transmission of Foot-and-Mouth disease virus in pigs. Vet. Res. 37(5); 647-54.

Orsel, K., de Jong, M. C. M., Bouma, A., Stegeman, J. A., Dekker, A. 2007. Foot and mouth disease virus transmission among vaccinated pigs after exposure to virus shedding pigs. Vaccine 25(34); 6381-6391.

Line 338: Do you mean that piglet were vaccinated at 80 days of age (11 weeks) then none of the nursery pigs would have had a vaccination.

References:

In reference 9 the authors are not correctly presented due to a mistake at the journal. It should read:

Vosloo, W., Nguyen, T. T. H., Fosgate, G. T., Morris, M. J., Wang, J., Kim, V. P., Quach, V. N., Le, T. T. P., Dang, H., Tran, X. H., Vo, V. H., Le, T. Q. A., Mai, T. M. T., Le, T. V. Q., Ngo, T. L., Singanallur, B. N. 2015. Efficacy of a high potency O1 Manisa monovalent vaccine against heterologous challenge with a FMDV O Mya98 lineage virus in pigs 4 and 7 days post vaccination. Vaccine 33(24); 2778-2785.

Comments on language mistakes are included in the other comments.

Author Response

We appreciate your effort to review our manuscript. Please see the attachment for our response to your individual comments.

Reviewer 2 Report

Dear P.T. Authors,

I would like to express my great interest in the subject of this manuscript and impressive analyses you’ve performed, using the data from the FMD outbreak in Korea in 2014. The results of your study give answers to all the farm owners and field veterinarians struggling to protect their herd from FMD. In the current world, where universal animal trade and movement constantly threatens introduction of infectious diseases, with FMD as one of the biggest challenges, the data you provide are important and crucial in preparation of contingency plans – especially in a scenario where the novel viruses are involved, towards which standard vaccines may not work.

In spite of the excellent job you did, there is still a field for improvements, as listed below:

1.       line 77-83 should be removed;

2.       all the symbols used in the manuscript should be explained there – e.g. ?;

3.       References should be updated with recent publications and the number of references older than 10 year should be reduced to minimum.

Thank you for your cooperation!

Author Response

(The authors gave the same response as above.)
